# Study on the Damage Model of Non-Persistent Jointed Rock Mass under the Coupling of Freeze–Thaw and Shear

**DOI:** 10.3390/ma16083041

**Published:** 2023-04-12

**Authors:** Daxing Lei, Haixiang Hu, Yifan Chen, Hang Lin, Chaomei Zhang, Guangli Wang, Zhigang Lu, Yaoping Zhang

**Affiliations:** 1School of Resources and Architectural Engineering, Gannan University of Science and Technology, Ganzhou 341001, China; daxinglei17@163.com (D.L.); rhrx@foxmail.com (C.Z.); zhiganglu14@163.com (Z.L.); yaopingzhang74@163.com (Y.Z.); 2Key Laboratory of Mine Geological Disaster Prevention and Control and Ecological Restoration, Ganzhou 341000, China; 3Chongyi Zhangyuan Tungsten Co., Ltd., Ganzhou 341300, China; 4School of Resources and Safety Engineering, Central South University, Changsha 410083, China; 15216122359@163.com (Y.C.); linhangabc@126.com (H.L.)

**Keywords:** jointed rock mass, freeze–thaw cycle, shear failure, mesoscopic damage, coupling damage

## Abstract

Considering that a jointed rock mass in a cold area is often affected by periodic freeze–thaw cycles and shear failure, definitions for the mesoscopic and macroscopic damage to a jointed rock mass under the coupling of freeze–thaw and shear are proposed, and the damage mechanism is verified according to experimental results. The results show that: (1) the jointed rock specimens increase macro-joints and meso-defects, the mechanical properties deteriorate significantly under freeze–thaw cycles, and the damage degree becomes more and more significant with the increases in freeze–thaw cycles and joint persistency. (2) When the number of freeze–thaw cycles is constant, the total damage variable value gradually increases with the increase in joint persistency. The damage variable difference in specimens with different persistency is distinct, which is gradually reduced in the later cycles, indicating a weakening influence of persistency on the total damage variable. (3) The shear resistance of non-persistent jointed rock mass in a cold area is determined by the coupling effect of meso-damage and frost heaving macro-damage. The coupling damage variable can accurately describe the damage variation law of jointed rock mass under freeze–thaw cycles and shear load.

## 1. Introduction

With the continuous expansion of mineral resource development in cold areas, it has been a common phenomenon that rock masses in opencast mining are affected by freeze–thaw cycles and shear load. For example, in northeast China, many jointed rock mass projects have obvious freeze–thaw damage and shear failure effects under a freeze–thaw environment, as shown in Figure 1. So, it is of great significance to study the shear failure law and damage development mechanism of jointed rock masses under freeze–thaw cycles [1,2,3,4,5,6].

Until now, multitudinous scholars have conducted effective research in this field. For example, many scholars evaluated the overall impact of the environment based on the shear creep failure of rocks under complex conditions and the evolution law of rock failure [7,8,9,10]. According to the feedback results, the experimental process and analysis method were optimized [11,12,13,14,15], and advanced experimental methods and perfect analysis theory were proposed [16,17,18,19,20]. Then, in order to measure and efficiently estimate the mechanical properties of rock materials under various conditions, many scholars used acoustic waves in real-time and non-destructive detection and monitoring applications to study the mechanical strength and macro- and microstructural characteristics of rock [21]. In addition, acoustic frequency analysis methods are also applied to study rock characteristics, and the results show that there is a reliable mathematical relationship between the two factors [22].

Zhang and Shen [23,24,25,26] derived the damage constitutive relationship at the meso-level when a jointed rock experienced freeze–thaw cycles by taking advantage of the heterogeneity and the Weibull distribution, on which basis the damage constitutive model of rock under the coupling action of freeze–thaw and conventional triaxial compression was established. Tan [27] applied the damage theory to analyze the correlation between the freeze–thaw damage and strain softening in rock. Li [28] studied the evolution of mechanical properties of jointed rock masses under freeze–thaw conditions and obtained damage models under various complex environments. Meanwhile, Yang [29] derived the damage model of rock under thermal–mechanical coupling from the mesoscopic perspective based on the internal microstructure characteristics of jointed rocks. Xu [30] and Kang [31] derived the strain-based damage evolution equation of rock under load using the Lemaitre strain equivalence principle and deduced the meso-damage model of rock with the coupling damage mechanism of freeze–thaw cycles and compression load. Based on experimental results, Bayram [32] established the strength damage evolution model of rock under the coupling action of freeze–thaw cycles and compression, considering the meso-damage evolution process. Meanwhile, Liu et al. [33,34] derived the freeze–thaw and load damage coupled mathematical formula of rock using the damage coupling relation, the applicability of which was verified with the experimental data. Lu et al. [35,36] conducted freeze–thaw cycle tests on fractured rock masses and studied the damage deterioration mechanism. Further, combined with the damage theory and macro-damage characteristics of rock masses, a damage model of freeze–thaw fractured rock mass under uniaxial compression was established. For opencast mining, however, there are a large number of non-persistent jointed rock masses. The non-persistent jointed rock masses are prone to localized damage and fracture failure along the joint after being affected by freeze–thaw cycles, which is essentially the failure problem caused by the repeated frost-heaving force. Furthermore, the external shear load will also cause substantial damage and deterioration to the jointed rock masses [37,38], significantly threatening the overall stability of rock mass in cold regions [39,40].

The research on rock damage is indeed of great significance to fully understand the mechanism of strength deterioration in jointed rock masses under freeze–thaw cycles and load, and its stability is difficult to predict accurately. Therefore, jointed rock masses have the hidden danger of geological disaster. However, most of the current strength deterioration in jointed rock masses was analyzed separately using damage at two different scales, while the damage characteristics of jointed rock masses under the coupling action of freeze–thaw cycles and shear are rarely reported. As two types of loads with great differences, the damage and fracture-inducing mechanism and evolution characteristics are completely different. Therefore, the research on the failure mechanism of jointed rock masses under the coupling mechanism of the freeze–thaw cycle and shear load will provide scientific evidence for the evolution of shear resistance characteristics of jointed rock masses and long-term safety protections in cold areas.

In this paper, considering the macro- and meso-coupling effects of freeze–thaw and shear loads, a damage model under the coupling effects of freeze–thaw and shear loads is established to describe the macro-damage performance of jointed rock mass and the mechanical property variation. In addition, the influences of freeze–thaw cycles and strain on the damage characteristics are analyzed, and the damage model is verified using the results of shear experiments on freeze–thaw rock, which presents satisfactory agreements.

## 2. Meso-Damage Model of Rock under Freeze–Thaw Cycle and Shear

### 2.1. Evolution of Freeze–Thaw Damage to Mesoscopic Defects in Rocks

According to Li, et al.’s [41,42,43] conclusions, the damage to different rocks in multiple freeze–thaw cycles can be described and analyzed from two perspectives. One perspective is that, under the condition of a single freeze–thaw cycle, the meso-defects inside the rock will cause structural damage during the freeze–thaw process. This is specifically manifested as the crystal structure inside the rock experiencing a series of damage processes including freezing, expansion, deterioration, thawing, and shrinkage, and the corresponding damage laws of each stage are quite different. Another perspective is that the internal structure of the rock is deteriorated for a long time due to the multi-freeze–thaw cycles. The meso-structure and defect distribution inside the rock are generally random, so there are many factors influencing the damage degree under the freeze–thaw cycle, and its evolution law is also complicated. Furthermore, the commonly used method is to measure the macroscopic mechanical properties before and after the freeze–thaw to quantitatively analyze the damage and deterioration degree in a rock. Therefore, the macroscopic damage mechanics method is one of the effective ways to analyze the mesoscopic damage degree of rock under freeze–thaw cycles. Moreover, in the process of freeze–thaw cycles, the shear modulus of a rock specimen can accurately show the variation in shear characteristics, so the freeze–thaw damage variable of rock λ can be defined as [4]:(1)$$D_{n}=1-\frac{G_{n}}{G_{0}}$$
where $D_{n}$ is the damage variable corresponding to the shear modulus after the freeze–thaw, $G_{0}$ is the shear modulus before the freeze–thaw cycle, and $G_{n}$ is the effective shear modulus after *n* freeze–thaw cycles.

According to the shear experiment results from a freeze–thaw specimen investigated by Lei [44], with the increase in freeze–thaw cycles, the peak shear strength, the corresponding peak displacement, and the slope of the linear variation section in the stress displacement curve of rocks will change. Similarly, the shear modulus also changes with the accumulation of freeze–thaw cycles. Therefore, the mesoscopic damage evolution law can be deduced based on the variation law of rock shear modulus *G*.

According to the calculation method for the shear modulus proposed by Tao [45], in common shear experiments, the shear stress is equal to 90% of the failure load, and the elastic structure of the shear plane is not obviously damaged. Therefore, based on the theory of material mechanics, it can be concluded that:(2)$$U=\frac{\tau^{2}}{2G}$$
where *U* is the rock shear strain energy and *G* is the rock shear modulus.

The shear stress–shear displacement curve (τ−s) can be plotted according to the rock shear experiment, so the strain energy *U* value under different shear stresses can be calculated. In addition, shear displacement needs to be converted into unit shear displacement, namely ε = *s*/*L* (*s* is the shear displacement and *L* is the length of the rock specimen along the shear direction). According to Equation (2), it can be deduced that:(3)$$G=\frac{\tau^{2}}{2U}$$

Furthermore, the Hooke law is satisfied before the shear failure of rock structures. Therefore, the formula can be derived as follows:(4)τ=Gε
where ε is shear strain.

Then, by substituting Equation (4) into Equation (3), the shear strain energy of rock can be expressed as:(5)$$U=\frac{\tau \varepsilon }{2}$$

In addition, during the shear process of rock, shear strain can be calculated according to the thickness of the shear band and the relative shear displacement. The principle of meso-structure is shown in Figure 2, and the thickness of the shear band basically remains constant with the increase in shear stress. Therefore, the calculation formula for strain can be expressed as [46]:(6)$$\varepsilon =\frac{\Delta S}{\Delta h}$$
where ΔS is the relative shear displacement during the experiment and Δh is the thickness of the shear band.

Therefore, according to the freeze–thaw shear experimental results in Lei [44], and combined with Equations (3)–(5), the shear modulus of rock specimens under different freeze–thaw cycles can be calculated, shown in Table 1.

Furthermore, the damage variable values of intact rock specimens under different freeze–thaw cycles can be calculated by substituting the shear modulus results into Equation (1), and the results are shown in Table 2.

It can be concluded that with increasing freeze–thaw cycles, the degree of freeze–thaw damage to intact specimens also increases. At the early stage of the freeze–thaw cycle, the freeze–thaw damage variable of the intact specimens increased significantly. Moreover, the freeze–thaw damage variable increased by 0.4 in the interval of 0–10 cycles and increased by 0.2 in the interval of 10–20 cycles, with a growth rate of 12% compared with the previous stage. However, the damage variables only increased by 5.0% and 4.6%, indicating that with the accumulation of freeze–thaw cycles, the growth trend in the damage variables of intact specimens gradually weakened and eventually leveled off.

### 2.2. Shear Damage Evolution of Rock Meso-Defects

Rocks contain various randomly distributed pores and mesoscopic defects, which can be regarded as the initial damage to rocks. Therefore, the damage degree inside the rock can be quantified using the strength of the element based on the principle of statistical mechanics, and the statistical damage constitutive model of the rock based on the initial damage state can be established according to the random distribution characteristics of the damage inside the rock. Further analysis shows that the damage variation in rock during shear is a continuous process, and the mechanical properties of rock elements can be quantitatively analyzed using the statistical method under the condition of a small defect scale. Therefore, it can be assumed that rock element strength follows the Weibull distribution, and its probability density equation is [47,48]:(7)$$P(\varepsilon )=\frac{m}{F_{0}}{\left(\frac{\varepsilon }{F_{0}}\right)}^{m-1}\exp\left[-{\left(\frac{\varepsilon }{F_{0}}\right)}^{m}\right]$$
where P(ε) is the probability density function corresponding to the rock element strength, ε is the rock micro-strain under shear load, and *m* and *F*_0_ are the function distribution parameters, representing the brittleness parameter and the macroscopic average strength of the rock, respectively.

In the process of the shear experiment, as the shear load on the rock continues to increase, when the shear strain reaches a certain threshold, the total number *n* of shear-damaged elements in the rock is [49,50,51]:(8)$$n=\int_{0}^{\varepsilon }NP\left(x\right)dx=N\left[1-e^{-{\left(\frac{\varepsilon }{F_{0}}\right)}^{m}}\right]$$
where *N* is the initial number of elements in the rock specimen.

In addition, the rock damage variable can be defined as *D*. The ratio between the total number of damaged elements *n* and the initial total number *N* of rock elements for rock specimens under a continuous shear load ranges from 0 to 1. Therefore, *D* can be expressed as:(9)$$D=\frac{n}{N}$$

Then, by combining Equations (8) and (9), the damage evolution equation for rock specimens under continuous shear load can be expressed as:(10)$$D=1-e^{-{\left(\frac{\varepsilon }{F_{0}}\right)}^{m}}$$

Meanwhile, according to the Hooke law, before the shear failure of intact rock, Equation (10) can be expressed as:(11)$$\frac{\tau }{G\varepsilon }=e^{-{\left(\frac{\varepsilon }{F_{0}}\right)}^{m}}$$

Then, logarithms are taken from both sides of the equation in Equation (11), and the transformation form can be evolved into:(12)$$\ln\left(-\ln\frac{\tau }{G\varepsilon }\right)=m\left(\ln\varepsilon -\ln F_{0}\right)$$

Therefore, the parameters *m* and $F_{0}$ of the probability distribution function can be fitted according to the data from the shear experiment. Moreover, according to the experiments carried out by Lei [44], *G* = 4.67 GPa is calculated when the rock specimen is not subjected to freeze–thaw. Combined with Equations (10) and (12) and the shear stress–strain theoretical curve, the comparison effect is close to that of the shear damage evolution curve of an intact rock specimen, respectively, as shown in Figure 3. Then, the fitting results show that the distribution function parameters *m* = 2.36 and $F_{0}$ = 0.074.

According to Figure 3a, the theoretical damage curve based on the Weibull distribution is basically consistent with the trend of the rock shear stress–strain curve before failure, and the peak failure strength is basically equal. The peak shear strength derived from the theory is almost consistent with the experimental results, while the strain corresponding to the peak strength is slightly larger than that in the theoretical model. In this paper, the meso-shear damage model of the intact rock specimen is simplified, and the influence of the initial tiny cracks and pores on the curve is ignored. Figure 3b shows that the mesoscopic damage is still at a low level before the shear peak strain, and the theoretical stress–strain curve presents an approximately linear variation. When the curve passed the peak strain point, the shear damage variable *D* increased sharply, while the stress–strain curve tended to moderate, and the shear stress gradually decreased, indicating that the shear rock entered the yield stage in this stage. Therefore, the meso-damage variable corresponding to the peak strain at the peak shear point can be regarded as the benchmark criterion for reaching the critical point of damage and failure in the shear process of rock.

### 2.3. Meso-Damage Evolution Equation for Rock under Freeze–Thaw and Shear Coupling

Under the influences of freeze–thaw cycles and shear, two different kinds of loads will cause damage and deterioration in rock specimens to different degrees, and the internal fine defects and pores will continue to develop and penetrate. Therefore, shear mesoscopic damage to a rock under the freeze–thaw cycles can be equivalent to the damage to a rock under the coupling of two kinds of loads. According to the Lemaitre equivalent strain principle and the concept of effective stress, it is known that the constitutive relation form of the material in any damaged state is the same [52]. Meanwhile, combined with the extended strain equivalent principle proposed by Yang [53], the initial state of the rock before freeze–thaw is taken as the first primary damage state, and the damage after freeze–thaw is taken as the second additional damage state. Then, it can be derived that:(13)$$\tau_{0}A_{0}=\tau_{n}A_{n}$$
(14)$$D_{n}=\frac{A_{0}-A_{n}}{A_{0}}$$
where $\tau_{0}$ and $\tau_{n}$ are the effective stress under the initial primary state and freeze–thaw cycles state, respectively, and $A_{0}$ and $A_{n}$ are the effective bearing area with shear under the initial primary state and freeze–thaw cycles state, respectively.

Then, combined with (13) and (14), the Equation can be expressed as:(15)$$\tau_{n}=\frac{\tau_{0}}{1-D_{n}}$$

In addition, the strain under various states can be transformed into:(16)$$\varepsilon =\frac{\tau_{0}}{G_{n}}=\frac{\tau_{n}}{G_{0}}$$

By combining Equations (15) and (16), the relation of the shear modulus under freeze–thaw cycles and the constitutive relation of the linear stage of freeze–thaw damage and failure of the rock can be expressed as:(17)$$G_{n}=G_{0}\left(1-D_{n}\right)$$
(18)$$\tau_{n}=G_{0}\left(1-D_{n}\right)\varepsilon_{n}$$

Meanwhile, the shear damage state of the rock after freeze–thaw is analyzed, and the state after the freeze–thaw damage is defined as the first damage state, which is consistent with Equation (1) and used to describe the parameters of freeze–thaw damage. The total shear damage after freeze–thaw is defined as the second damage state, and by applying the strain equivalence principle after extension, the constitutive relation of shear damage after freeze–thaw can be expressed as:(19)$$\tau =G_{n}\left(1-D\right)\varepsilon$$
where D is the mesoscopic damage variable in the shear process.

Combined with Equations (17) and (19), the stress–strain relationship of the shear rock after freeze–thaw in a linear stage is converted into:(20)$$\tau =G_{0}\left(1-D_{t}\right)\varepsilon$$

The expression of the total damage to the rock $D_{t}$ can be expressed as:(21)$$D_{t}=D_{n}+D-D_{n}D$$

According to Equation (21), under the combined action of freeze–thaw and shear, the total mesoscopic damage variable of the rock keeps increasing, with a typical non-linear growth trend. Freeze–thaw damage is essentially the damage caused by the micro-pores inside the rock and the frost-heaving force inside the macro-joint. Shear loading destroys the micro-pores inside the rock and the pore structure inside the joint, which leads to slip and dislocation and finally causes the coupling damage of freeze–thaw damage and shear, which aggravates the total damage degree of the rock.

Combined with Equations (7), (10), (18) and (21), the total damage evolution equation of rock under freeze–thaw shear coupling is:(22)$$D_{t}=1-\frac{G_{n}}{G_{0}}e^{-{\left(\frac{\varepsilon }{F_{0}}\right)}^{m}}$$

According to Equation (22), when the rock specimen is only damaged by the freeze–thaw cycle, it is not affected by shear load, and the shear strain is converted to ε=0, and combined with the Equation (21), the total damage variable is converted to $D_{t}=D_{n}$. Moreover, when the rock specimen is only subjected to shear load, the damage variable $D_{n}=0$ of the rock specimen under freeze–thaw, after substituting into Equation (17), the initial shear modulus is converted to $G_{0}=G_{n}$, and then the total damage variable is converted to $D_{t}=D$.

## 3. Damage Model of Jointed Rock Mass under Freeze–Thaw and Shear

### 3.1. Coupling Damage Variables of Jointed Rock Mass under Freeze–Thaw and Shear

According to the research results of Yan, et al. [54,55], the essence of damage coupling at different scales is still the coupling of multiple damage variables. The macroscopic cracks generated by frost heaving in the process of freeze–thaw are the result of the continuous evolution and development of microscopic micro-cracks [2]. Therefore, to analyze the damage evolution law of jointed rock mass under freeze–thaw and shear, it is necessary to calculate the macro- and mesoscopic damage variables of the jointed rock mass at different scales. The following basic assumptions can be adopted:(1)When analyzing the classification of the damage state of jointed rock mass, the damage inside the rock mass in the shear process and micropore frost heaving damage inside the rock mass in the freeze–thaw process are summarized as mesoscopic damage. The macroscopic joint defects and cracks caused by frost heaving in the rock mass are summarized as macroscopic damage, and the macroscopic frost heaving damage is evolved from the microscopic freeze–thaw damage.(2)Both macroscopic damage and mesoscopic damage to joint specimens in the freeze–thaw cycle and the shear loading process are homogeneous and isotropic.(3)Based on the strain equivalent principle proposed by Lemaitre, the damage coupling of a jointed rock mass during freeze–thaw cycles and the shear process is analyzed.

The condition of damage coupling caused by shear and freeze–thaw cycles is under a certain shear stress, and the sum of the strains caused by two kinds of damage is equal to the strains caused by the coupling damage at different levels. As shown in Figure 4, the states of rock specimens represented by definitions (a)–(d) are: rock specimens containing both macroscopic joints and mesoscopic pores, rock specimens containing only macroscopic joints, rock specimens containing only mesoscopic pores, and rock specimens without defects under ideal conditions, whose shear moduli are respectively: ${\bar{G}}_{12}$, ${\bar{G}}_{1}$, ${\bar{G}}_{2}$, and ${\bar{G}}_{0}$. The strains caused by shear stress τ are $\varepsilon_{12}$, $\varepsilon_{1}$, and $\varepsilon_{0}$, respectively. According to the principle of equivalent strain, it can be deduced that:(23)$$\varepsilon_{12}=\varepsilon_{1}+\varepsilon_{2}-\varepsilon_{0}$$

In Figure 4, the rock specimens in the four states satisfy the Hooke law under the shear stress τ, then the transformation form of the strain equation is:(24)$$\frac{\tau }{{\bar{G}}_{12}}=\frac{\tau }{{\bar{G}}_{1}}+\frac{\tau }{{\bar{G}}_{2}}-\frac{\tau }{{\bar{G}}_{0}}$$

According to the strain equivalence principle and damage theory hypothesis, the damage condition of the intact rock specimen in Figure 4d is taken as the basic damage state, then the macro-damage caused by the prefabricated macro-joint in the rock specimen in Figure 4b is defined as $D_{x}$. For the damage to the rock specimen in Figure 4c, the mesoscopic damage caused by the freeze–thaw cycle and the shear of intact rock without macro-damage is defined as $D_{y}$, while the sum of macro-mesoscopic damage to the prefabricated macro-jointed rock specimens under freeze–thaw and shear in Figure 4a is defined as $D_{xy}$. In addition, the relative size of the shear modulus represents the degree of damage deterioration in jointed rock specimens, then the shear modulus of each stage is expressed as:(25)$$\begin{array}{c} {\bar{G}}_{12}={\bar{G}}_{0}(1-D_{xy})\\ {\bar{G}}_{1}={\bar{G}}_{0}(1-D_{x})\\ {\bar{G}}_{2}={\bar{G}}_{0}(1-D_{y}) \end{array}\}$$

Meanwhile, by substituting Equation (25) into Equation (24), the total damage amount in freeze–thaw shear joint specimens can be converted into:(26)$$D_{xy}=1-\frac{\left(1-D_{x}\right)\left(1-D_{y}\right)}{1-D_{x}D_{y}}$$

According to Equation (26), $D_{xy}=D_{x}$ can be deduced when the rock mass contains only macro-joint defects, that is, $D_{y}=0$. Under this condition, the macro- and mesoscopic coupling damage variable of the jointed rock mass is equal to the macro-damage variable of the rock mass. If the rock mass does not contain macroscopic joints and other defects, that is, $D_{x}=0$, $D_{xy}=D_{y}$ can be converted, that is, the macro- and mesoscopic coupling damage variable of the rock mass is equal to the mesoscopic damage variable of the rock mass, which is consistent with the actual conditions. Therefore, the macro- and mesoscopic coupling damage variables of a jointed rock mass under freeze–thaw and shear established using the above theoretical methods are consistent with engineering practice.

Since perfectly ideal undamaged rocks do not exist, almost all rocks contain initial damage. Therefore, it is difficult to quasi-calculate the shear modulus of a rock without initial damage according to the ideal state. However, according to the relativity principle of the definition of damage variables, the initial state of intact rock can be defined as the reference damage state. Therefore, the damage to rock specimens with macroscopic joints is defined as $D_{c}$, while the total damage to intact rock caused by the freeze–thaw cycle and shear is defined as $D_{t}$, and the total damage to rock specimens with macroscopic joints by the freeze–thaw cycle and shear load is defined as $D_{m}$. Therefore, Equation (26) can be converted into:(27)$$D_{m}=1-\frac{\left(1-D_{c}\right)\left(1-D_{t}\right)}{1-D_{c}D_{t}}$$

In the expression of total damage $D_{m}$, $D_{c}$ can be expressed as:(28)$$D_{c}=1-\frac{G_{j}}{G_{c}}$$
where $G_{j}$ is the equivalent shear modulus of the jointed rock mass, $G_{c}$ is the shear modulus of the rock with complete damage state, and $D_{t}$ can be calculated using Equation (21).

### 3.2. Calculation of Macro- and Micro-Defect Damage Variables of Jointed Rock Specimens

A rock mass usually contains non-persistent joints with different persistency, so it is complicated to analyze the macro-defects in each persistency joint separately. Therefore, the non-persistent joints distributed in an engineering rock mass can be simplified as macro-defects, and the joints can be abstracted as macro-damage based on the initial state of intact rock with reference to the damage state. At present, many scholars first define damage variables according to the geometric parameters of joints and then introduce correction coefficients to correct the variables in the damage equation [56,57]. In addition, according to the strain energy release model proposed by JA [58], under shear stress τ for all kinds of jointed rock specimens, the macro- and micro-damage strain energy release rate is:(29)$$R=-\frac{\tau^{2}}{2G_{0}{\left(1-D_{xy}\right)}^{2}}$$
where $D_{xy}$ is consistent with Equation (26) in the meaning of damage.

Meanwhile, the expression of elastic strain energy per unit volume can be transformed into:(30)$$U^{G_{12}}=-\left(1-D_{xy}\right)R=\frac{\tau^{2}}{2G_{0}(1-D_{xy})}$$

Thus, when the rock specimen is not subjected to freeze–thaw cycles and does not contain macroscopic joints, $D_{x}=0$ and $D_{xy}=D_{y}$, and then the expression of strain energy $U^{G_{0}}$ per unit volume of the rock specimen at this time can be expressed as:(31)$$U^{G_{0}}=\frac{\tau^{2}}{2G_{0}\left(1-D_{y}\right)}$$

The expression of the difference in strain energy $\Delta U^{G}$ per unit volume of the rock specimen caused by joint damage to macroscopic joints and freeze–thaw cycles can be expressed as:(32)$$\Delta U^{G}=U^{G_{12}}-U^{G_{0}}$$

Substituting Equations (30) and (31) into Equation (32), the expression can be expressed as:(33)$$\Delta U^{G}=\frac{\tau^{2}}{2G_{0}}\left(\frac{1}{1-D_{xy}}-\frac{1}{1-D_{y}}\right)$$

Then, in combination with Equations (26) and (33), it can be deduced that:(34)$$\Delta U^{G}=\frac{\tau^{2}}{2G_{0}}\left(\frac{1}{1-D_{x}}-1\right)$$

Moreover, based on the conclusions of Mohammadi H [59] and J. Kemeny [60] and the principle of fracture mechanics, the difference in elastic strain energy accumulated by joints in unit volume of a rock specimen is formulated as:(35)$$\Delta U^{G}=\rho_{v}\int_{0}^{A}G^{\prime }dA$$
where *A* is the joint surface area, $\rho_{v}$ is the joint volume density, and $G^{\prime }$ is the energy release rate of the rock specimen.

Therefore, according to the physical and mechanical properties of rocks and joints under the coupling effects of freeze–thaw and shear, the difference in strain energy can be calculated.

Combined with Equations (31), (33) and (34), the total coupling damage variables of jointed rock specimens under freeze–thaw and shear can be calculated. According to the definition for each initial damage state of jointed rock samples, $D_{xy}$ is equivalent to the total damage $D_{m}$. Therefore, the theoretical calculation results based on Equations (34) and (35) can be combined with Equations (22), (27) and (28) to verify the damage model of jointed rock mass under freeze–thaw conditions and shear loads.

## 4. Verification and Analysis of Model Examples

In order to further analyze the applicability of the damage model, this paper verifies it according to Lei’s [44] shear test results. After 0, 10, 20, 30, and 40 freeze–thaw cycles, the rock specimens in each group of persistency are subjected to shear tests. The results show that the more freeze–thaw cycles, or the higher the persistency of joints, the more significant the damage to the specimen. The shear modulus data for the joint specimens under different freeze–thaw cycles are shown in Table 3.

When analyzing the freeze–thaw degree of a jointed rock mass, the damage variable value of the stress peak point can be used as the reference value to measure the freeze–thaw damage degree of the rock mass. According to Equation (5), the meso-damage variable *D* of intact rock under shear can be calculated. When ε is the peak strain, the initial damage to the intact rock sample *D* = 0.072, and the damage $D_{n}$ after freeze–thaw of the rock specimen can be calculated using Equation (1). Meanwhile, the damage $D_{t}$ to the joint specimen under the coupling of freeze–thaw and shear can be calculated using Equation (22). In addition, the total damage $D_{m}$ to the joint specimen under the coupling of freeze–thaw and shear macro- and meso-damage can be calculated using Equations (27), (28), (34) and (35).

Based on the shear experiment results and Equation (28), the variation curves for the macro-damage variables of joint specimens with different persistency can be plotted, as shown in Figure 5a. Using Equation (1), the freeze–thaw damage variable $D_{n}$ evolution curve of the complete rock specimen can be calculated, as shown in Figure 5b. It is found from Figure 5a that the macro-damage variable of the joint specimen increases linearly with the increase in persistency. When the persistency of the joint is equal to 20%, the damage variable $D_{c}$ is 0.24, while when the persistency of the joint is equal to 40%, the damage variable $D_{c}$ is 0.46. Furthermore, the damage variable is increased by 91.6%, indicating that the damage degree of the rock caused by the increase in the persistency of the macro-joint is particularly significant. It can be concluded from Figure 5b that the damage variable of the complete rock specimen presents an exponential rising trend with the increase in the freeze–thaw cycles. The damage variable increases greatly in the initial stage of freeze–thaw cycles, and then gradually tends to moderate in the later stage. For example, when the freeze–thaw cycle *n* = 10, 20, and 30, the damage variable $D_{n}$ is 0.13, 0.19, and 0.23, respectively, and the growth rate of the damage variable decreases gradually. The reason for the above results is that repeated freeze–thaw has caused a large degree of damage to the joint rock specimen. In the late freeze–thaw cycles, the proportion of water migration is larger, and the frost-heaving effect is gradually weakened, so the growth rate of damage variables also gradually decreases.

The freeze–thaw joint specimens are divided into three groups according to the persistency, and the joint distribution is shown in Figure 6. According to the above derivations, the theoretical values of the total damage variable $D_{m}$ of a jointed rock specimen under freeze–thaw and shear can be calculated by combining Equations (31), (33)–(35). According to the shear test results, the experimental value of the total damage variable $D_{m}$ can be calculated by substituting them into Equations (22), (27) and (28).

Figure 7 shows the variation trend in the calculated results and test results of the total damage variable $D_{m}$ for joint specimens under the coupling effects of freeze–thaw cycles and shear loads. It gradually increases with the increase in persistency and presents an upward convex increase with the increase in freeze–thaw cycles. When the freeze–thaw cycles are constant, the total damage variable $D_{m}$ also increases gradually with the increase in joint persistency. In the early stage of freeze–thaw cycles, the difference in the damage variable of specimens with different persistency is large, which shows a decreasing trend from freeze–thaw cycles to 40 times, indicating that the increase in joint persistency in the late stage of freeze–thaw cycles has a gradually weakened influence on the total damage variable. The total damage variables of each joint specimen all show a maximum increase during the 0–10 cycles, and the increased amplitude tended to moderate during the 10–20 cycles. Finally, the total damage variables basically maintained a slightly increasing trend during the 30–40 cycles. These results show that the total damage degree is the highest in the first 10 freeze–thaw cycles, and the cumulative effect of the overall damage degree is gradually weakened after 20 freeze–thaw cycles.

For example, when the persistency is 20% and the joint specimen freeze–thaw cycle is 0, 10, 20, 30, and 40 times, the total damage variable $D_{m}$ is 0.22, 0.33, 0.42, 0.47, and 0.49, respectively, while the theoretical calculation results are 0.20, 0.3, 0.41, 0.47, and 0.51, respectively, as shown in Figure 7a. During the 0–10 freeze–thaw times, the total damage variable is increased by 50.0%, and it is increased by 27.2%, 11.9%, and 4.68%, respectively, during the 10–20, 20–30, and 30–40 freeze–thaw stages. It can be concluded that the total damage variable increased significantly in the early stage of the freeze–thaw cycle, while it is very weak in the late stage. Therefore, macro-damage to the joint specimen leads to an aggravation trend in macro- and micro-damage under shear load. After the freeze–thaw cycle, the internal structure of the joint rock specimen deteriorated significantly under the frost-heaving force of multiple cycles, which leads to the gradual weakening of shear mechanical properties. Finally, the freeze–thaw jointed rock specimen stimulates a certain degree of macroscopic damage weakening in the shear process.

Furthermore, the variation trend in $D_{m}$ calculated values and test values for the three groups of samples is basically consistent. This presents a great agreement and indicates that the coupling calculation method based on the macro- and meso-damage variables proposed in this paper is feasible and can accurately describe the damage variation in a joint rock mass under the freeze–thaw cycle and shear.

## 5. Conclusions

Rock masses in cold regions are exposed to damages caused by freeze–thaw cycles and shear loads, which can be simplified as the coupling damage to jointed rock masses under freeze–thaw cycles and shear. Through quantitatively characterizing the coupling relationship between freeze–thaw damage and shear damage in jointed rock samples, the evolution law between the mesoscopic damage state and macroscopic mechanical response of a jointed rock can be finally revealed. The research conclusions are as follows:(1)The jointed rock sample contains macroscopic joints and microscopic defects, and its mechanical properties will deteriorate under the action of the freeze–thaw cycle and shear load. The more freeze–thaw cycle times or the higher persistency of the joint, the heavier the damage degree of the sample. When the persistency is 20% and the joint specimen freeze–thaw cycle is 0, 10, 20, 30, and 40 times, the total damage variable is 0.22, 0.33, 0.42, 0.47, and 0.49, respectively.(2)For the same freeze–thaw cycles, the total damage variable under the coupling effect of freeze–thaw and shear gradually increases with the increase in joint persistency. In the early stages of freeze–thaw cycles, the difference in damage variables with different persistency is large, which shows a decreasing trend for the 40 cycles, indicating that the influence of joint persistency on the total damage variables gradually weakens in the late freeze–thaw cycles.(3)According to the proposed coupling damage model, it is found that the test results for the total damage variables are in good agreement with the analysis conclusions. The variation trend in the tested and theoretical total damage variables with the increase in freeze–thaw cycles is basically the same. This shows that the coupling damage model of jointed rock mass established according to the macro- and meso-damage variables can accurately reflect the damage variation law of rock mass under the freeze–thaw cycle and shear action.

## Figures and Tables

**Figure 1 materials-16-03041-f001:**
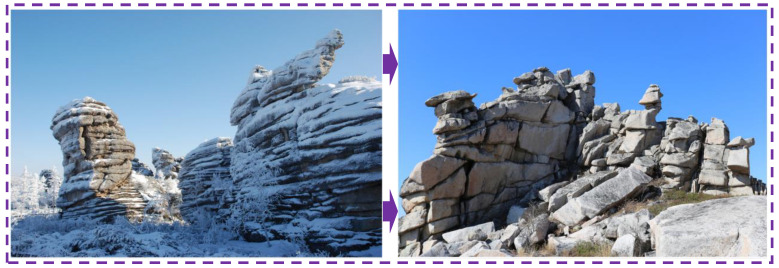
Macroscopic damage and deterioration of jointed rock masses under freeze–thaw shear in northeast China.

**Figure 2 materials-16-03041-f002:**
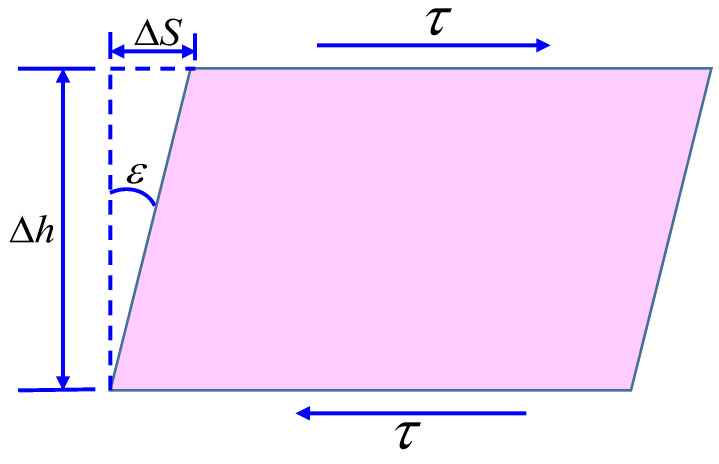
Schematic diagram showing the force analysis of rock shear zone [45].

**Figure 3 materials-16-03041-f003:**
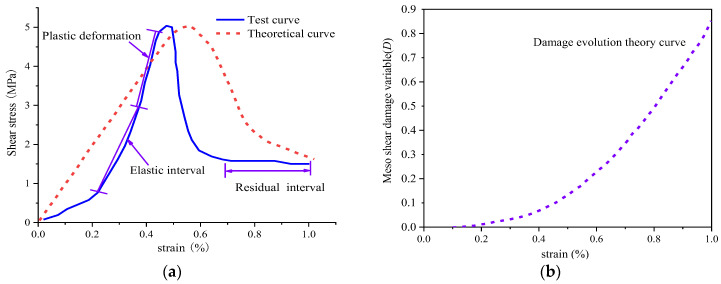
Complete rock specimen shear damage model evolution curve for the (**a**) comparison between the theoretical and experimental stress–strain and (**b**) meso shear damage theory [45,48,49].

**Figure 4 materials-16-03041-f004:**
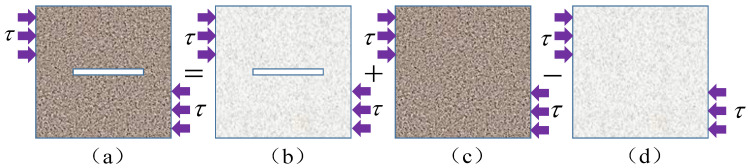
(**a**) Joint sample damaged by freeze-thaw and shear coupling, (**b**) joint sample with shear damage, (**c**) intact rock samples with freeze-thaw damage, (**d**) intact rock sample with shear damage. Calculation method for the equivalent strain of jointed rock specimens.

**Figure 5 materials-16-03041-f005:**
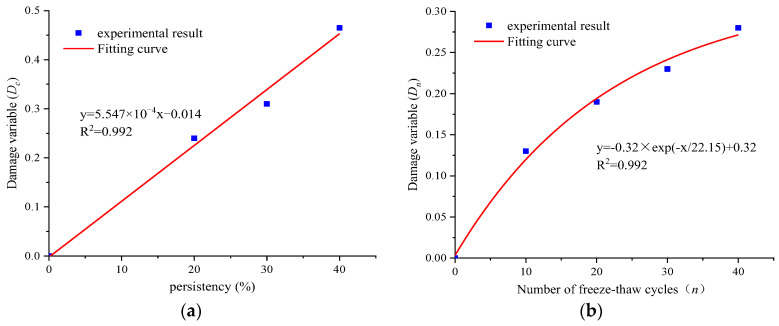
Damage evolution curve of (**a**) a jointed rock with persistency and (**b**) an intact rock with freeze–thaw cycles.

**Figure 6 materials-16-03041-f006:**
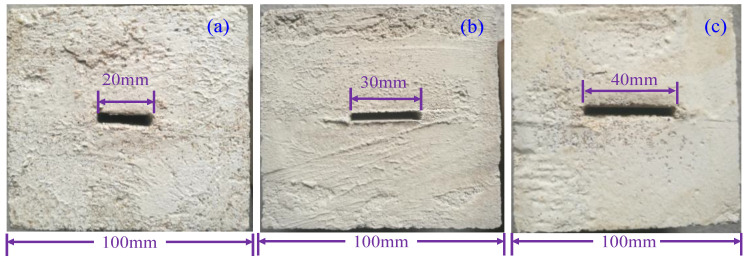
Joint distribution form of freeze–thaw specimens with (**a**) persistency = 20%, (**b**) persistency = 30%, and (**c**) persistency = 40%.

**Figure 7 materials-16-03041-f007:**
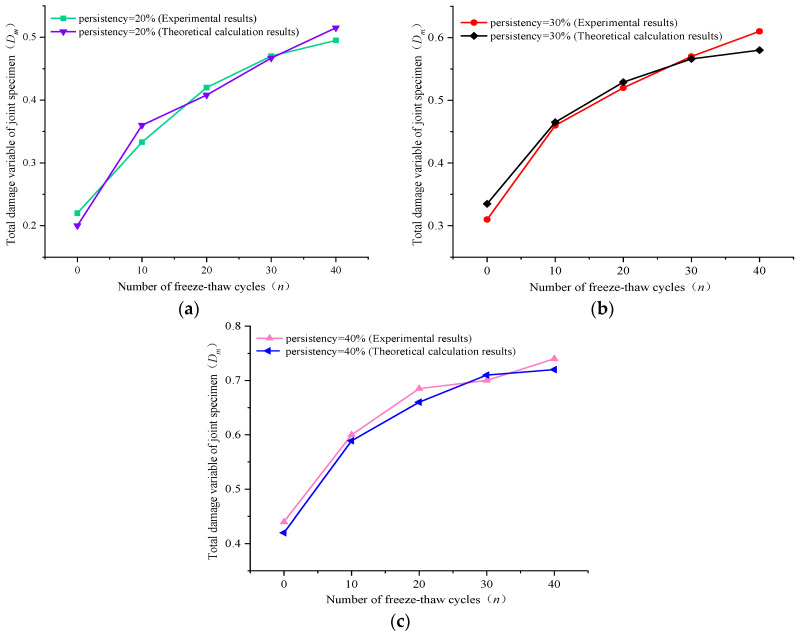
Comparison between the calculation and experimental results for the total damage variables in freeze–thaw shear of (**a**) persistency = 20% joint specimens, (**b**) persistency = 30% joint specimens, and (**c**) persistency = 40% joint specimens.

**Table 1 materials-16-03041-t001:** Variation characteristics of the shear modulus with the freeze–thaw cycles.

Specimen Characteristic	Shear Modulus/GPa				
	0 Freeze–Thaw Cycles	10 Freeze–Thaw Cycles	20 Freeze–Thaw Cycles	30 Freeze–Thaw Cycles	40 Freeze–Thaw Cycles
Intact rock specimen	4.67	4.01	3.61	3.30	2.96

**Table 2 materials-16-03041-t002:** Damage variable of the shear modulus with freeze–thaw cycles.

Specimen Characteristic	$\mathrm{Damage}\;\mathrm{Variable}/D_{n}$				
	0 Freeze–Thaw Cycles	10 Freeze–Thaw Cycles	20 Freeze–Thaw Cycles	30 Freeze–Thaw Cycles	40 Freeze–Thaw Cycles
Intact rock specimen	0.072	0.091	0.11	0.20	0.32

**Table 3 materials-16-03041-t003:** Variation in the shear modulus of joint specimens with freeze–thaw cycles.

Freeze–Thaw Cycles (*n*)	Shear Modulus (GPa)			
	0	10	20	30
0	4.67	3.82	3.53	2.86
10	4.01	3.16	2.67	1.88
20	3.61	2.64	2.04	1.10
30	3.30	2.18	1.47	0.71
40	2.96	1.82	1.08	0.55

## Data Availability

Some or all data, models, and code that support the findings of this study are available from the corresponding author upon reasonable request.

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
