# Peer review of "Study on the Damage Model of Non-Persistent Jointed Rock Mass under the Coupling of Freeze–Thaw and Shear"

_materials, 2023, doi:10.3390/ma16083041_

Round 1

Reviewer 1 Report

Dear Authors,

The paper entitled “Study on the damage model of non-persistent jointed rock mass under the coupling of freeze-thaw and shear” is considered accurately. The topic is so interesting and well prepared. The paper could be accepted in Materials after editing/answering following major revision comments/questions:

1. What are the scientific reasons of authors for selecting this method of research? I think it was better to publish diverse parts of this research in separate paper!

2. Numbers of references could increase for an original research paper using researches published by diverse studies carried out whole world, not in certain region.

3. References are checked completely; it is suitable and covers old and novel research.

4. Is this research is based on a real case study or it is just an abstract modeling? The geological map of case study area which samples are gathered and its location containing its scale could help to better understanding of readers about condition of case study.

5. The graphical quality of Figures 1 and 3 should enhanced.

6. For a comprehensive review in introduction part, in Introduction part authors should mention to diverse methods of measuring and measuring and estimating of strength of rigid materials like concrete/rocks. Also, novel methods like acoustic frequency analysis for estimating the materials are eliminated in introduction. These issues must be added to introduction. Following references could be helpful for this purpose and authors should use them in introduction part for increasing comprehensiveness of this part:

7. Investigating the acoustic signs of different rock types based on the values of acoustic signal RMS; M Khoshouei, R Bagherpour, MH Jalalian, M Yari; Rudarsko-geološkonaftni zbornik (The Mining-Geological-Petroleum Bulletin)

8. Developing a novel model for predicting geomechanical features of carbonate rocks based on acoustic frequency processing during drilling; M Yari, R Bagherpour, M Khoshouei; Bulletin of Engineering Geology and the Environment 78 (3), 1747-1759

9. The paper is too long for a research paper. As a suggestion, authors could summarize some parts of paper.

Author Response

Reviewer 1

The paper entitled “Study on the damage model of non-persistent jointed rock mass under the coupling of freeze-thaw and shear” is considered accurately. The topic is so interesting and well prepared. The paper could be accepted in Materials after editing/answering following major revision comments/questions:

1 What are the scientific reasons of authors for selecting this method of research? I think it was better to publish diverse parts of this research in separate paper!

Response: Thank you for your overall positive comment, In the previous freeze-thaw experiments, the deterioration trend of shear strength of jointed rock mass was obtained. However, the coupling effect of freeze-thaw damage and shear failure, which are two different forms of damage, needs to be further studied to accurately predict the shear strength of jointed rock mass. Therefore, the coupling damage model at macro and micro level is obtained by theoretical derivation, and verified based on the experimental results, which shows the validity of the derivation results. The relevant content has been refined in the revised manuscript.

2 Numbers of references could increase for an original research paper using researches published by diverse studies carried out whole world, not in certain region.

Response : Thank you very much for your suggestion. The relevant references have been added in the revised manuscript.

3 References are checked completely; it is suitable and covers old and novel research.

Response : Thank you for your careful review. We have carefully scrutinized the manuscript, and we have revised the relevant references.

4 Is this research is based on a real case study or it is just an abstract modeling? The geological map of case study area which samples are gathered and its location containing its scale could help to better understanding of readers about condition of case study.

Response :Thanks for your suggestions, We have added the relevant content.

This study is based on a real engineering background, and the object of this study is the failure of jointed rock mass in northeast China under the coupling effects of freeze-thaw and shear under the large diurnal temperature difference. The experimental results verify the feasibility of the coupled damage model.

And the revised content is as follows:

“For example, in northeast China, many jointed rock mass projects have obvious freeze-thaw damage and shear failure effects under freeze-thaw environment.”

Fig.1 Macroscopic damage and deterioration of jointed rock mass under freeze-thaw shear in northeast China

  1. The graphical quality of Figures 1 and 3 should enhanced.

Response: Thanks for your suggestions, We have revised the Fig 1 and 3 to address your concerns and hope to meet your approval.

Fig.2 Schematic diagram of force analysis of rock shear zone

Fig.4 Calculation method of equivalent strain of jointed rock specimens

  1. For a comprehensive review in introduction part, in Introduction part authors should mention to diverse methods of measuring and measuring and estimating of strength of rigid materials like concrete/rocks. Also, novel methods like acoustic frequency analysis for estimating the materials are eliminated in introduction. These issues must be added to introduction. Following references could be helpful for this purpose and authors should use them in introduction part for increasing comprehensiveness of this part:

Response: Thank you for your careful review. We have carefully scrutinized the manuscript, and the revised content is as follows:

Until now, multitudinous scholars have conducted effective research in this field. For instance, In order to measure and estimate efficiently the mechanical properties of rock materials under various conditions, many scholars have used acoustic waves in real-time and non-destructive detection and monitoring applications to study the mechanical strength and macro and micro structural characteristics of rock[21]. In addition, the acoustic frequencies analysis methods are also applied to research rock characteristics, the results show that there is a reliable mathematical relationship between the two factors [22].

  1. Investigating the acoustic signs of different rock types based on the values of acoustic signal RMS; M Khoshouei, R Bagherpour, MH Jalalian, M Yari; Rudarsko-geološkonaftni zbornik (The Mining-Geological-Petroleum Bulletin)

Response: Thank you for your suggestion. We have studied the contents of this paper and found it very valuable and have quoted the views of the paper.

  1. Developing a novel model for predicting geomechanical features of carbonate rocks based on acoustic frequency processing during drilling; M Yari, R Bagherpour, M Khoshouei; Bulletin of Engineering Geology and the Environment 78 (3), 1747-1759

Response: Thank you for your suggestion. We very much agree with the research point of the paper and have quoted it in the revised manuscript

  1. The paper is too long for a research paper. As a suggestion, authors could summarize some parts of paper.

Response: Thank you for your overall positive comment, the relevant content has been refined and removed, and the format has been carefully reviewed and corrected, the manuscript has been streamlined and optimized.

Finally, we would like to express our acknowledgements to the anonymous reviewers for their precious comments. Thanks are also given to the editors during the submission of our manuscript.

Best regards,

Sincerely yours,

Daxing Lei

GanNan University of Science And Technology, Ganzhou, Jiangxi, 341000, China

Reviewer 2 Report

The paper covers key aspects of freeze-thaw cycles and shear loading affecting open pit mines in cold regions. It is crucial to understand the mechanisms of freeze-thaw and shear loading and their effect on the behaviour of the rockmass. The authors derive a mathematical model which can be used to predict the behaviour of the rock mass over time as it is exposed to cycles of freeze-thaw and shear loading. The mathematical model is calibrated using experimental results from freeze-thaw specimens. The experimental rock specimens have macroscopic joints and microscopic defects. Their mechanical properties were moted to deteriorate under the action of freeze-thaw cycle and shear load. The research noted that the more freeze-thaw cycle times or the higher the persistency of joint, the heavier the damage degree of the sample.

 While this is a good technical paper backed with both mathematical derivations and laboratory experiments, it would have been more valuable if the authors had made practical observations at an operational open pit mine in a cold region. They would make practical monitoring of freeze-thaw cycles and shear loading, recording the resultant displacements and deformation. This would definitely improve the predictive accuracy of their mathematical model in evaluating practical rockmass behaviour when exposed to freeze-thaw action and shear loading.

Author Response

Reviewer 2

The paper covers key aspects of freeze-thaw cycles and shear loading affecting open pit mines in cold regions. It is crucial to understand the mechanisms of freeze-thaw and shear loading and their effect on the behaviour of the rock mass. The authors derive a mathematical model which can be used to predict the behaviour of the rock mass over time as it is exposed to cycles of freeze-thaw and shear loading. The mathematical model is calibrated using experimental results from freeze-thaw specimens. The experimental rock specimens have macroscopic joints and microscopic defects. Their mechanical properties were moted to deteriorate under the action of freeze-thaw cycle and shear load. The research noted that the more freeze-thaw cycle times or the higher the persistency of joint, the heavier the damage degree of the sample.

While this is a good technical paper backed with both mathematical derivations and laboratory experiments, it would have been more valuable if the authors had made practical observations at an operational open pit mine in a cold region. They would make practical monitoring of freeze-thaw cycles and shear loading, recording the resultant displacements and deformation. This would definitely improve the predictive accuracy of their mathematical model in evaluating practical rock mass behaviour when exposed to freeze-thaw action and shear loading.

Response: Thank you very much for your suggestion. The background of our research project is the damage of jointed rock mass after freezing and thawing in open-pit mines in cold regions. The preliminary content of the project is model experiment and comparative analysis based on numerical simulation. Finally, the relevant theoretical analysis is carried out, and the corresponding strength criteria and mathematical relations are derived.

After the completion of the preliminary theoretical research results, we will focus on the actual production mines as the basis of analysis of the mine's actual freeze-thaw environment and damage deterioration law, as well as the evolution characteristics of slope shear failure, according to the field investigation conclusions for engineering verification, with a large number of practical engineering cases to support the research conclusions, and in the relevant mines to promote the conclusion. 

Finally, we would like to express our acknowledgements to the anonymous reviewers for their precious comments. Thanks are also given to the editors during the submission of our manuscript.

Best regards,

Sincerely yours,

Daxing Lei

GanNan University of Science And Technology, Ganzhou, Jiangxi, 341000, China

Reviewer 3 Report

See the submitted file.

Author Response

Responses to Reviewers’ comments

Dear Editor and Dear Reviewers:

Thank you for your valuable comments to our manuscript, titled with “Study on the damage model of non-persistent jointed rock mass under the coupling of freeze-thaw and shear”, which is submitted to the journal of “materials”. According to your comments, we have made corresponding revisions for the manuscript, the revisions parts have been marked by red color. Attached are the responses to the comments:

Reviewer 3

This paper uses the damage theory to simulate the degradation of mechanical properties by subjecting them to freeze-thaw cycles. In the suggested articles there are papers that have been used or can be used for this proble. The paper is based on a known methodology and the problem studied has been considered in other papers so maybe it is not so original but that does not mean that it is not interesting. The methodology is also supported by experimental data which shows very good agreement with the theoretical one, of course, the experiment is done with the existence of a crack which is easy to model. The text is well-written, and the conclusions are supported by the results.

I have only few suggestions to make for improving your manuscript and mainly consist of adding reference that had exclude from your research:

Point 1: In the initial stages of loading the moduli is different because the rock is deformed due to open porosity, then when the grains come into contact, they give a different slope without any degradation of properties. The model that approximates this concept is that of Walsh:

[1] https://doi.org/10.1029/JZ070i002p00399.

Response 1: Thank you for your overall positive comment and suggestion, we have added relevant citations.

There are indeed some differences between the theoretical and experimental models in this manuscript in the initial loading stage, and Walsh model is more consistent with the changes in the modulus of rocks after loading. In addition, this study should not only consider the change law of shear modulus of rocks under load, but also analyze the peak shear strength of rocks after freeze-thaw cycles. Therefore, this manuscript refers to the classical shear stress-strain curve and focuses on peak correlation, and then introduces the overall damage variable.

[49] Walsh J B . The effect of cracks on the uniaxial elastic compression of rocks. Journal of Geophysical Research, 1965, 70(2):399-411.

Point 2: In strain energy of line 405 you can refer and Kemeny and Cook model using this parameter to define effective moduli:

[2] https://doi.org/10.1016/0148-9062(86)90337-2

Response 2: Thank you for your suggestion. The literature you pointed out is really valuable. We have added the literature citations to the manuscript. According to the original equation derivation process, we think that equation (35) can verify the total damage variable in equation (27).

[60] J. Kemeny, N.G.W. Cook. Effective moduli, non-linear deformation and strength of a cracked elastic solid. International Journal of Rock Mechanics & Mining Sciences & Geomechanics Abstracts, 1986, 23(5):107–118.

Point 3: Also, read this relevant article for application of fracture mechanics for propagation and interaction of cracks under freeze-thaw cycling in rock-like material

[3] https://doi.org/10.1016/j.ijrmms.2022.105112

Response 3: Thank you for your careful review and patient guidance, we have revised the content of the manuscript and added references.

[3] Tang X ,  Tao S ,  Li P , et al. The propagation and interaction of cracks under freeze-thaw cycling in rock-like material. International Journal of Rock Mechanics and Mining Sciences, 2022(154-):154.

Point 4: Why two captions exist on Figure 6, Figure 6-Figure 7. When in one figure consists of 2 or more figures the explanation go on the figures caption.

Response 4: Thank you for your careful review. We have carefully reformatted Figure 6 and Figure 7, adjusted the position order, and revised the manuscript content.

Finally, we would like to express our acknowledgements to the anonymous reviewers for their precious comments. Thanks are also given to the editors during the submission of our manuscript.

Best regards,

Sincerely yours,

Daxing Lei

GanNan University of Science And Technology, Ganzhou, Jiangxi, 341000, China

Reviewer 4 Report

·        Line 21: change to “increase”

·        Along theory has been explained in the abstract. The author should limit the abstract to introduction, methodology, results, and conclusion.

·        Space should be given between text and reference.

·        Line 52. Change to “was proposed”

·        Line 64: “Therefore” to be deleted

·        Author may cite this article https://doi.org/10.3390/app12052752

·        Clear significance/problem statement of research should be added in the manuscript.

·        Line 108: instead of writing “previous conclusions”, it is suggested to mention the Author’s names.

·        Better to give reference of EQ 1.

·        It is suggested to cite Fig. 2 first then Fig.3.

·        Marking stages in Figure 2(a,b) will give better understanding to the reader.

·        Justify the statement given between lines 293-297 by giving explanation or reference.

·        Line 3.2: instead of writing “research conclusions of literatures”, it is suggested to mention the Author’s names.

·        What are the basis of assumptions made between lines 310-319.

·        Line 431-436: it is suggested to write the calculated values of Dn, Dt, and Dm,

·        Fig. 7: How much is the difference between experimental and theoretical results?

·        Conclusion (1) is well known and general.

·        It is suggested to include quantitative analysis results as well.

Author Response

Responses to Reviewers’ comments

Dear Editor and Dear Reviewers:

Thank you for your valuable comments to our manuscript, titled with “Study on the damage model of non-persistent jointed rock mass under the coupling of freeze-thaw and shear”, which is submitted to the journal of “materials”. According to your comments, we have made corresponding revisions for the manuscript, the revisions parts have been marked by red color. Attached are the responses to the comments:

Reviewer 4

Point 1:  Line 21: change to “increase”

Response 1: Thank you for pointing this out. We have revised the relevant content.

Point 2:  Along theory has been explained in the abstract. The author should limit the abstract to introduction, methodology, results, and conclusion.

Response 2: Thanks for your suggestion. We have simplified and corrected the abstract,

The relevant revisions are as follows:

    The definitions of mesoscopic and macroscopic damages of jointed rock mass under the coupling of freeze-thaw and shear are proposed, and the damage mechanism is verified according to the experimental results. The results show that: (1) the jointed rock specimens increase macro-joints and meso-defects, the mechanical properties deteriorate significantly under freeze-thaw cycles, and the damage degree becomes more and more significant with the increases of freeze-thaw cycles and joint persistency. (2) When the number of freeze-thaw cycles is constant, the total damage variable value gradually increases with the increase of joint persistency. The damage variable difference of specimens with different persistency is distinct, which is gradually reduced in the later cycles, indicating a weakening influence of persistency on the total damage variable.

Point 3:  Space should be given between text and reference.

Response 3: Thank you very much for your careful review, we have revised it.

Point 4:    Line 52. Change to “was proposed”

Response 4: Thanks for your advice, we have corrected the error.

Point 5:  Line 64: “Therefore” to be deleted

Response 5: Thanks for your suggestion, we have revised it.

Point 6:  Author may cite this article https://doi.org/10.3390/app12052752

Response 6: Thanks for your suggestion, we have added relevant literature.

[6] Umer Waqas, Hafiz Muhammad Awais Rashid, Muhammad Farooq Ahmed. Damage Characteristics of Thermally Deteriorated Carbonate Rocks: A Review. Appl. Sci. 2022, 12, 2752.

Point 7:  Clear significance/problem statement of research should be added in the manuscript.

Response 7: Thank you for your sincere opinion, we have refined the research significance and engineering problems in the manuscript.

Point 8:     Line 108: instead of writing “previous conclusions”, it is suggested to mention the Author’s names.

Response 8: This suggestion is great, we have revised the content of the manuscript.

Point 9:  Better to give reference of EQ 1.

Response 9: Thank you for your suggestion. We have added references to the manuscript.

Point 10:  It is suggested to cite Fig. 2 first then Fig.3.

Response 10: That's a great idea, We have adjusted the reference.

Point 11: Marking stages in Figure 2(a,b) will give better understanding to the reader.

Response 11: Thank you for your suggestion. We have added phase annotations to the Figure.

(a)                                      (b)

Fig. 3 Complete rock specimen shear damage model evolution curve of (a) Comparison of theoretical and experimental stress-strain , (b) meso shear damage theory[45,48,49]

Point 12: Justify the statement given between lines 293-297 by giving explanation or reference.

Response 12: Thank you, your suggestion is very valuable. We have added relevant explanations to the manuscript.

 Point 13:  Line 3.2: instead of writing “research conclusions of literatures”, it is suggested to mention the Author’s names.

Response 13: Thank you for your careful review, we have revised the relevant content of the manuscript.

 Point 14: What are the basis of assumptions made between lines 310-319.

Response 14: Thank you for pointing this out. We have added references to the manuscript.

According to the scanning results of Reference 2, it is found that the freeze-thaw damage defects inside the rock samples are all microscopic cracks at the beginning. When the number of freeze-thaw cycles is large, the microscopic cracks will gradually evolve into macroscopic cracks. In the shearing process, the macroscopic crack is also gradually evolved from the microscopic crack.

 Point 15:  Line 431-436: it is suggested to write the calculated values of Dn, Dt, and Dm,

Response 15: Thank you for your suggestion. The freeze-thaw conditions and macro-rock cracks in this paragraph have not been determined, therefore, the values of damage variables will be calculated in the following manuscript

In this paragraph of the manuscript,  ,and  are only introduced that numerical values can be calculated by using multiple equations simultaneously. At this time, the number of freeze-thaw cycles has not been determined, so a definite value cannot be given directly. When the number of freeze-thaw cycles n=0, the values of  ,and are 0, 0.072, 0.081 respectively.

Point 16:  Fig. 7: How much is the difference between experimental and theoretical results?

Response 16: Thank you very much for your careful review.

For example, when the persistency is 20% and the joint specimen freeze-thaw cycle are 0, 10, 20, 30, 40 times, the total damage variable is 0.22, 0.33, 0.42, 0.47, 0.49, respectively. While the theoretical calculation results is 0.20, 0.30, 0.41, 0.47and 0.51, respectively. The theoretical and experimental differences are 9%, 9%, 2%, 0% and 4%, respectively.

Point 17:  Conclusion (1) is well known and general. It is suggested to include quantitative analysis results as well.

Response 17: Your opinion is very valuable, we have revised and supplemented Conclusion (1).

Finally, we would like to express our acknowledgements to the anonymous reviewers for their precious comments. Thanks are also given to the editors during the submission of our manuscript.

Best regards,

Sincerely yours,
Daxing Lei

GanNan University of Science And Technology, Ganzhou, Jiangxi, 341000, China

Round 2

Reviewer 1 Report

the paper could be published in present form.

Author Response

Thank you for your excellent work and positive comments